# Protein Fold Usages in Ribosomes: Another Glance to the Past

**DOI:** 10.3390/ijms25168806

**Published:** 2024-08-13

**Authors:** Inzhu Tanoz, Youri Timsit

**Affiliations:** 1Aix-Marseille Université, Université de Toulon, IRD, CNRS, Mediterranean Institute of Oceanography (MIO), UM 110, 13288 Marseille, France; inzhu.tanoz@etu.univ-amu.fr; 2Research Federation for the Study of Global Ocean Systems Ecology and Evolution, FR2022/Tara GOSEE, 3 Rue Michel-Ange, 75016 Paris, France

**Keywords:** protein fold, ribosome evolution, super-ribosome folds, molecular convergence, power-law distribution, RNA-protein co-evolution

## Abstract

The analysis of protein fold usage, similar to codon usage, offers profound insights into the evolution of biological systems and the origins of modern proteomes. While previous studies have examined fold distribution in modern genomes, our study focuses on the comparative distribution and usage of protein folds in ribosomes across bacteria, archaea, and eukaryotes. We identify the prevalence of certain ‘super-ribosome folds,’ such as the OB fold in bacteria and the SH3 domain in archaea and eukaryotes. The observed protein fold distribution in the ribosomes announces the future power-law distribution where only a few folds are highly prevalent, and most are rare. Additionally, we highlight the presence of three copies of proto-Rossmann folds in ribosomes across all kingdoms, showing its ancient and fundamental role in ribosomal structure and function. Our study also explores early mechanisms of molecular convergence, where different protein folds bind equivalent ribosomal RNA structures in ribosomes across different kingdoms. This comparative analysis enhances our understanding of ribosomal evolution, particularly the distinct evolutionary paths of the large and small subunits, and underscores the complex interplay between RNA and protein components in the transition from the RNA world to modern cellular life. Transcending the concept of folds also makes it possible to group a large number of ribosomal proteins into five categories of urfolds or metafolds, which could attest to their ancestral character and common origins. This work also demonstrates that the gradual acquisition of extensions by simple but ordered folds constitutes an inexorable evolutionary mechanism. This observation supports the idea that simple but structured ribosomal proteins preceded the development of their disordered extensions.

## 1. Introduction


*Ce texte cite une «certaine encyclopédie chinoise » où il est écrit que «Les animaux se divisent en: (a) appartenant à l’Empereur, (b) embaumés, (c) apprivoisés, (d) cochons de lait, (e) sirènes, (f) fabuleux, (g) chiens en libertés, (h) inclus dans la présente classification, (i) qui s’agitent comme des fous, (j) innombrables, (k) dessinés avec un pinceau très fin en poil de chameau, (l) et caetera, (m) qui viennent de casser la cruche, (n) qui de loin semblent des mouches» (Borges). Préface de «les mots et les choses» Michel Foucault.*


### 1.1. About Usages in Complex Systems

The usage or preferred usage of the elements that compose a complex system can reveal valuable information about its nature, the laws that govern it, its evolution, or the respective functions of each of its components. Certain systems that are the fruit of different evolutionary processes, such as the universe, life, human languages, or scientific literature, present analogies in the occurrence of their constituents. For example, far from being homogeneous, the use of words in a language, references in scientific papers, or codons or folds in genomes are biased and often form part of what is known as a power law or Zipf’s law [1,2,3,4,5,6,7]. However, these analogies may derive from different evolutionary mechanisms and organisational principles of their own. In linguistics, the use of characters, words, or ideograms can, for example, reveal the grammatical principles governing the language and its evolution. The preferential use of synonyms can also provide interesting information About the semantic features, the author style or the text puropose’s [8,9,10,11]. Biological systems are also characterised by biases in the use of certain elements, which not only reveal their properties but can also provide clues to their history. For example, in modern bacteria, the bias of riboswitches towards some ligands may be evidence of systems that existed in the RNA world [12]. Another phenomenon—codon usage bias—is the preferential use of certain synonymous codons in genomes, genes, or even different gene regions [13,14]. Growing evidence shows that codon usage may modulate the speed of protein biosynthesis in the ribosome. The resulting non-uniform ribosome local kinetics—denoted as translational rhythm—modulate the translation rates and co-translational protein folding of the ribosome [15,16]. Fold usage in structural databases and extant genomes revealed several important features about the laws that govern the protein universe [3,17]. These studies revealed that the distribution of folds also follows a power law, and that there is, therefore, a minority of folds—the superfolds—that are extremely frequent in databases and genomes, such as P-loop NTPases, Rossmann-folds, ferredoxin-like domains, and TIM-barrels [4,18,19,20,21,22,23,24,25]. However, many factors affect the distribution of folds, which, for example, differs according to kingdoms, species, or an organisms’ lifestyles. For example, the vibrational energy exchanges among protein residues [26] and allostery [27] have been shown to correlate with protein topology. Protein dynamics vary according to their fold [28], so that each has its own dynamic personality [29]. On the other hand, alpha helices are more robust to mutations than beta strands, [30] and some folds are more likely to evolve and develop new functions than others [31]. Fold usage can therefore provide functional and evolutionary insights as well as information on a fold’s ancestral character, bearing in mind that superfolds are considered to be very ancient folds, which would have predated the radiation of the three kingdoms [32,33,34,35,36].

The analysis of fold usage has not yet, to our knowledge, been extensively applied to the functional and evolutionary study of ribosomes, the organelles responsible for protein biosynthesis. The ribosomes are ribonucleoprotein particles (roughly one-third protein and two-thirds RNA for cytoplasmic ribosomes that are composed of a large (LSU) and a small (SSU) subunit). Ribosomes perform translation—mRNA coded protein synthesis—in the three domains of life [37,38,39,40,41,42,43]. Since the 2000s, Solving the ribosome structures from the three kingdoms structures from the three kingdoms, with or without their various substrates and factors, has revealed not only the breathtaking details of the mechanisms of each stage of translation (initiation, elongation, termination, and recycling), but also how their complexity has increased over the course of evolution. Thus, the ribosome is an ideal system for studying evolutionary transitions because it constitutes, in particular its peptidyl-transfer center, a ‘window to the past’ [44,45,46,47]. Thought to be an autonomous system before the radiation of the three kingdoms, it may represent a precious missing link in evolution [48,49,50,51,52]. According to various hypotheses, its structure could be used to infer the “first steps” in the evolution of a peptide/RNA world. First of all, ribosomes are a model of choice for understanding the mechanisms and evolution of the concerted assembly of nucleoprotein complexes [53,54]. From a functional point of view, a ribosome is also an extremely complex system that carries out protein biosynthesis. Over the last 20 years, structural studies have made it possible to follow the various stages and dynamics of translation at the angstrom scale [55,56]. These studies have revealed the details of an extremely complex dynamic that underpins the fidelity of protein biosynthesis. These studies also highlighted the synchronization of translation players and communication between the functional sites. Ribosomes are also considered relics of the transition from the RNA world to the first peptide/RNA life forms [57,58,59], reflecting the co-evolution of translation players and the genetic code [60,61,62]. Consisting of an interesting repertoire of RNA modules [63,64] and ancestral proteins, they could reflect the first forms of organization and the primordial mutualism of these two categories of polymers [65]. As such, they would be at the crossroads of the evolutionary processes that probably preceded the last universal common ancestor (LUCA) [66] at the end of the so-called “RNA world”. Numerous studies converge on the idea that ribosomes evolved by accretion and that their constituents have their own evolutionary scenarios [52]. Ribosomal RNA (rRNA) and the ribosomal proteins that make up the two subunits would thus have had different origins and would then have co-evolved to form ‘modern’ ribosomes [67,68,69]. An important component of ribosomes that play an essential role in their assembly and translation are ribosomal proteins [70,71,72,73].

Ribosomal proteins are unique in that they are composed of a globular domain, usually anchored to the surface, and long, disordered, filamentous extensions that thread their way inside the crevices formed by the rRNA [74]. The role of these extensions is still enigmatic, but numerous studies converge on the idea that they play multiple roles. For example, they may participate in the assembly of the large [75] or small subunit [76]. However, some experimental studies have shown that they can both play a role in assembly [77] and participate in very specific tasks during translation [78], such as allosteric coordination between functional sites [79]. The evolution and phylogeny of ribosomal proteins [80,81] has been documented by numerous studies, showing that they have followed complex evolutionary scenarios in different kingdoms. In addition, based on studies of the gradual evolution of protein motifs [82,83], it has been proposed that ribosomal proteins evolved from their disordered extensions because they contact deep, ancient regions of the rRNA [84]. Other studies have shown that ribosomal protein folds subsequently acquired these extensions, which weave increasingly complex networks as they pass from prokaryotes to eukaryotes [85,86,87]. These networks are thought to be involved in communication between distant functional parts of the ribosome and to synchronize increasingly complex tasks in the course of evolution [88].

To complement existing studies on the evolution and functions of ribosomal proteins, our short paper examines the distribution and usage of protein folds in ribosomes across different kingdoms. While it has been noted that ribosomes contain proteins that adopt very ancient folds, a detailed comparative analysis of protein fold distribution in the ribosomes of the three kingdoms has not, to our knowledge, been carried out. In the light of studies showing how, for example, Small Beta Barrel domains tolerate insertions, decorations, and other variations [89], and knowing that segment insertion and deletion (INDELS) plays an important role in protein evolution [90,91,92,93,94], we also examine how the different ribosomal protein folds acquire extensions. Are there different categories of extensions associated with specific folds? Finally, comparing the use of folds in ribosomes and in extant genomes provides interesting information for assessing differences in selective pressures at different stages of evolution.

### 1.2. “To Be or Not to Be” a Protein Fold

While we still wonder about the meaning of Shakespeare’s famous “to be or not to be” [95], the biophysics of protein folds [96] could well be engaged in such metaphysical reflection today, seeking their place and meaning in the complex landscape of evolution [97]. The protein fold has been a very useful conceptual unit that has enlightened both structural biology and evolution. However, while ‘structure–function relationships’ have shaped the landscape of molecular biology for almost 60 years, a number of studies have gradually shaken the immutability of the notion of protein’s structure and topology on one side and the concept of function on the other side. “To be or not to be” a protein fold is an important question to examine before analysing “fold usage” in genomes and in ribosome.

A fold is a biological entity that is the product of a spontaneous process, the hierarchical folding of a polypeptide chain into secondary and tertiary structures. Several decades of experimental and theoretical works have converged on the idea that proteins cooperatively fold to their native conformation, driven by negative ∆*G*, on funnel-shaped energy landscapes in the microsecond or millisecond time range [98,99,100,101]. The rate-limiting event in the folding reaction is the formation of a transition-state whose topology is determined by a set of interactions involving a small number of key residues and is close to that of the native state. Thus, protein sequences encode the information needed in both the hydrophobic folding nucleus and surface to guide the protein-folding reaction and generate the native fold [102,103,104,105,106]. However, a recent work propose an alternative view in which protein folding may follow a wide variety of rugged and dynamic landscapes, resulting in a broad range of thermodynamic and kinetic stability [107]. In addition, while some proteins are capable of folding on their own, others rely on molecular chaperones which are crucial for the folding and assembly of many multi-domain and multi-subunit proteins [108,109,110].

The origins of the protein fold classification seem to go back to the 1970s and 80s, when the first attempts emerged to categorise protein domains by structural similarity [111,112]. Hierarchical classification systems then progressively counted more than a thousand folds classified according to their contents in secondary structures and their topologies [113,114,115]. However, as seen in the famous “encyclopédie chinoise” presented by Michel Foucault (see above), the desire to classify living organisms very quickly came up against serious conceptual obstacles. Although very fruitful in the development of structural biology, fold categorizations have gradually found it difficult to adapt to the changing nature and complexity of macromolecules. Protein fold classifications soon came up against numerous pitfalls. One of them is, for example, the discovery that many proteins or protein regions do not adopt a structure and that some of them can only fold in the presence of partners [116]. These discoveries have led to a gradual recognition of the role of disorder in protein functionality, and intrinsically unstructured proteins have become one of the new paradigms of molecular biology in recent decades [117]. Another surprise in the same vein was the discovery by molecular dynamics studies that a significant proportion of folds or domains do not behave as autonomous folding units [118]. So, paradoxically, “not to be” a domain is therefore a relatively common property of protein folds. To complicate matters a little further, it was realised quite early on that the same sequence could alternatively adopt a beta-sheet or alpha-helix structures [119,120]. These switch phenomena, firstly localised in small protein sequences, were subsequently found to be much more frequent and able to extend to large regions or even entire proteins [121,122,123,124]. “Structural disorder” has been, indeed, found to be the physiological and functional state for most IDPs/IDRs. To be disordered or not to be disordered: is that still a question for proteins in the cell? [125]. Interestingly, the disorder content seems to increase during evolution, reaching up to 33% in eukaryotic proteomes. Thus, an increase in disordered regions in proteins seems to be associated with new signaling functions present in eukaryotes [126,127,128,129,130] and taking into account the cellular context [131].

### 1.3. Folding or Misfolding: That’s a Question of Ribosome?

An important step was then taken with the discovery of two different folding states of the same ribosomal protein, bL20, trapped within a single crystal [132]. This study has demonstrated experimentally how the interplay of structural context and electrostatics could influence protein folding. Figure 1a,b also show how the two forms, the partially unfolded form and the native form, stabilize each other in the crystal packing, in the manner of chaperones. Later, numerous studies showed that many “metamorphic” or “transformer” proteins can reversibly adopt distinct folds [133,134,135,136,137,138,139,140]. Among these chameleon-like sequences [141], one category—prions—has attracted particular attention because the conformational conversion of their cellular form (PrPc) into the pathologic isoform (PrPSc) has been found to cause fatal neurodegenerative diseases such as Bovine spongiform encephalopathy, scrapie of sheep, and Creutzfeldt–Jakob disease. Prions were the first example of transmissible particles composed exclusively of a modified protein (PrPSc) that are devoid of nucleic acid [142,143]. Prions have been found in the neurons and glyal cells of mammals and birds and also have homologs in yeast, as well as containing a disordered N-terminal region and a helical N-terminal domain [144,145]. The conversion of the α-helices of the normal functional (PrPc) protein into β-sheets initiates a self-perpetuating aggregation process in prion diseases [146,147,148,149,150,151,152]. Interestingly, the ribosome and RNA play a central role in misfolding processes and the onset of these diseases [153,154,155,156,157].

Beyond the metaphysical aspect of oscillating between “being or not being” a protein fold, folds have been also “transcended” into *urfolds* [158], *metafolds* [159], or “concepts”, [160] entities that considerably broaden the conceptualization of the protein structure universe. Many structures, such as, for example, small β-barrels (SBB), display striking analogies in their properties that “transcend” the concept of folds, allowing to group slightly different folds into a single category. Small beta barrels which include OB-folds and SH3 domains [89] have remarkably similar properties, probably reflecting a common and ancient origin [161]. Along with β-propeller blades [162] and large families, which includes, for example, P-loop NTPases and Rossmann-folds [33,163,164], protein structures can be viewed as entities that transcend the current topological classification [165]. 

In addition, the discovery of enzyme promiscuity [166], that a folds such as the TIM-barrel can be associated with a large number of functions [167], and, conversely, that different, non-homologous folds have evolved convergently to catalyse similar biochemical reactions [168] defies the classical paradigm of “structure–function relationships”. Convergent evolution at the molecular level is also illustrated by the independent evolution of propulsion systems for cell motility in the three kingdoms [169]. The reception and emission of light in bioluminescent organisms is also the result of fascinating convergent evolutionary mechanisms [170]. After the “not to be”, the long journey of fold metaphysics also takes us into the realms of “Je est un autre” (“I is another”) (A. Rimbaud), where molecular mimicry [171,172,173,174] makes us realize that evolution has played “imitation games”, such as protein folds being interchangeable. Convergent evolution can achieve feats of imitation that make a protein and a nucleic acid resemble each other, even to the point of luring in molecular recognition processes [175,176]. A spectacular illustration of this phenomenon of molecular mimicry is the virtually equivalent ring shape formed by the protein chains of opposite polarity of MutS and topoisomerase II [177] (Figure 1c,d). Interestingly, most of these mechanisms were already present before the radiation of the three kingdoms, and have helped shape the evolution of the ribosomal system.

## 2. Results

### 2.1. Ribosomal Protein Fold Distribution

#### 2.1.1. Overview

In the ribosomes of bacteria, archaea, and eukarya, the most prevalent protein fold class is αβ (approximately 50%), followed by mainly β (approximately 25%) and then mainly α (approximately 17%) (Figure 2a,b). A similar protein fold class distribution with mixed folds belonging to the αβ class, being the most prevalent followed by mainly β and then mainly α, was observed in modern genomes [4]. Remarkably, this reveals continuity between ribosomes and modern genomes, as the proportions of the secondary structure content have remained consistent. The prevalence of αβ-class folds aligns with the findings of Winstanley et al. [36], who assert that folds in the αβ class are more ancient. Additionally, the most common architecture in the ribosomes of the three kingdoms is the two-layer sandwich. In terms of protein fold occurrence, ribosomes exhibit a pattern where only a few topologies are prevalent, while the majority occurs just once (Figure 2c,d). The prevalent folds that occur in the ribosomes two and more times are super ribosome folds (SRF).

Comparison of the ribosomes of the three kingdoms also provides evolutionary insights. An interesting way of quantifying fold redundancy in a system is to calculate the protein/fold ratio. A ratio of 1, for example, indicates that, in a given system, each protein adopts a different fold. If the ratio increases, this indicates that several different proteins adopt the same fold. Interestingly, Table 1 shows that the protein/fold ratio increases from 1.13 to 1.42 from bacteria to eukaryotes, and that SSU still has a lower protein/fold ratio than LSU (Table 1). This trend is similar to that observed in modern genomes: as the genome size increases, the protein-to-fold ratio also increases [18]. Due to the limited number of different folds, the likelihood of discovering a new fold decreases with the increase in genome size in modern genomes and the number of nucleotides in ribosomes. In addition, different patterns of SRF distribution are observed in the three kingdom’s ribosomes. Bacterial ribosomes contain 52 proteins with 46 distinct topologies, with the OB fold and α − β Plaits being the most represented, occurring five and four times, respectively (Table 1). Archaeal ribosomes contain 65 proteins with 49 different folds where SH3 type barrels and the OB fold are the most prevalent, occurring six and five times, respectively. Eukaryotic ribosomes share the same proteins [178] and topologies as archaeal ribosomes, plus some additional ones, resulting in a total of 77 ribosomal proteins with 54 different topologies, with the SH3 type barrel and Arc Repressor Mutant fold being the most frequent, occurring seven and six times, respectively. However, we should note that, according to the recent finding of “ribosome heterogeneity”, slight variations in protein and fold content may also occur among species, among different strains of the same bacterial species, and even between different tissues within the same organism in multicellular organisms [179,180,181,182,183]. The Arc Repressor Mutant, subunit A, which is not a SRF in bacterial and archaeal ribosomes, becomes a SRF in eukaryotic ribosomes by being present in a third of the eukaryote-specific ribosomal proteins, and it remains a superfold in modern genomes. In contrast, other SRFs are more or less shared across the three kingdoms. Interestingly, the bacterial SRFs such as OB folds, α-β Plaits, Helix Hairpins, and SH3 type barrels have remained highly occurring folds and their percentages in modern genomes are likewise high [19]. In contrast, some archaeal and eukaryotic SRFs, such as the N-terminal domain of TfIIb and the UB roll, are not superfolds in current genomes. Conversely, some superfolds present in modern genomes, like the Rossmann fold, are not highly represented in ribosomes. 

#### 2.1.2. OB and SH3 Domains—The Small Barrel SRFs

Among the most prevalent folds across the three kingdoms are the OB fold and SH3-type barrels, both of which are small β-barrels. These compact, evolutionarily ancient, and functionally versatile protein domains are characterized by their small size (around 100 residues over five β-strands) and highly conserved structural framework [89]. The prevalence of an OB fold and SH3 domain in bacterial and eukaryotic ribosomes, respectively, believed to share a common ancestry [161], aligns with their crucial roles in fundamental biological processes such as transcription, translation, replication, and genome stability [185,186]. The SH3 and OB domains share significant structural similarities, with most residues in four β-strands occupying equivalent positions. Their common elements include the shared strand, the entire CS-sheet, the central strand of the NC-sheet, and a short loop of the NC-sheet, forming the SH3/OB common core. Variations in strand β5 include its absence in some OB domains, displacement in others, and distortion by additional β-strands in some SH3 domains. Other variable regions generally cause minor perturbations, except for notable cases such as uL24 (SH3) that contains a β hairpin, slightly deforming the CS-sheet [161]. 

It is interesting to note that the OB fold is overrepresented already in ribosomes due to its unique and versatile properties of RNA binding, and it is also prominent in modern genomes, serving various functions. The widespread presence of the OB fold in modern genomes highlights its two key characteristics: fold-related binding sites that evolution can adapt for diverse functions and the ability to bind a wide range of oligonucleotide sequences. The latter enables rapid and extensive diversification [187]. In contrast, proteins with the SH3 domain in extant organisms can bind to a variety of proline-rich motifs, participating in signal transduction and the regulation of enzymatic activity [188]. This suggests that ribosomal proteins containing the SH3 domain may share similar properties, supporting the idea of allosteric networks among ribosomal proteins [85] and increased connectivity among eukaryotic ribosomal proteins [87].

#### 2.1.3. The Premise of the Power-Law Behavior

In modern genomes, the distribution of protein folds follows a power-law pattern, with most folds being rare and only a few being highly occurrent [18]. While the number of genes increases exponentially, the diversity of protein folds does not expand at the same rate. When the number of folds in terms of percentages is plotted against the fold occurrences on a log–log graph, a straight line can be observed which indicates the presence of a power-law relationship. This power-law distribution of protein folds illustrates the evolutionary principle that a few highly efficient and adaptable folds can fulfill a wide array of functional roles, while numerous other folds may be more specialized or less common. 

Despite the R^2^ values on the log–log graphs (Figure 3a–c) not being very close to one, there are clear trends indicating the premises of a power-law relationship. We observe this power-law behavior across different ribosomes, and it becomes more pronounced when we examine the protein fold distributions from bacteria to eukarya (Figure 3d–f). Eukaryotic ribosomes contain more SRFs and they take up a bigger proportion compared to their bacterial and archaeal counterparts, with the prevalence of these SRFs progressively increasing from bacteria to eukarya. The protein-to-fold ratio also increases (Table 1), indicating that, with the emergence of archaea- and eukarya-specific proteins, the number of protein folds does not increase as significantly (Figure 4). This supports the idea that ribosomes might represent a crucial link between prebiotic chemistries and the emergence of the first genomes and cells. Thus, ribosomes, along with their translation factors and aminoacyl-tRNA synthetases, can be viewed as a primitive autonomous biological system [49].

Importantly, as mentioned earlier, the prevalent folds in ribosomes differ from those found in extant genomes. For instance, the Rossmann fold, which is one of the most prevalent superfolds in extant genomes due to its ability to accommodate a wide variety of catalytic functions, appears in the three kingdoms’ ribosomes only three times. More precisely, it is present in the form of minimal Rossmann-like motif (RLM). Both uS2, one of the domains of uL1 [189], and uL4 (see below in the section urfolds) contain β1, α1, and β2 elements followed by an additional helix and an additional strand ending with α2 and β3 elements [190]. 

#### 2.1.4. Different Fold Usages in the LSU and SSU

In the ribosomes of all three kingdoms, the general compositions of the large (LSU) and small (SSU) ribosomal subunits differ significantly due to the abundance of topologies specific to each ribosomal protein. This fits well with evolutionary studies suggesting that the two subunits evolved independently [52,67,68]. Although the exact timeline of their relative ages remains debated, it is believed that each subunit has distinct molecular origins. The evidence indicates that both subunits independently developed their RNA folding capabilities and ancestral functions before co-evolving to form the modern ribosome.

We noticed a bigger degree of differentiation between the LSU and SSU in archaeal and eukaryotic ribosomes in comparison to those in bacterial ones. Specifically, when comparing the LSU and SSU of bacterial ribosomes, it is evident that, while each ribosomal protein typically has distinct topologies, they share the same prevalent folds. Even though SH3-type barrels appear three times exclusively in the LSU and RP S5 domain 2 appears twice only in the SSU; some of the other most common folds, such as the OB fold and α-β Plaits, have similar occurrences in both subunits (Figure 4a). In contrast, in archaeal and eukaryotic ribosomes, the differences in fold distribution between the subunits are more pronounced (Figure 4b). For example, the SH3-type barrel appears six times in the LSU but only once in the SSU. Conversely, in the SSU, the OB fold is the most prevalent, occurring four times, while it appears only once in the LSU. This pattern supports their independent evolutionary paths and specialized functions within the ribosome.

### 2.2. Ribosomal Protein Folds Specific to Each Kingdom

#### 2.2.1. Minimal Fold Content

We also examined the folds present in the ribosomes of the three kingdoms to determine the minimal fold content (Appendix A). While most ancestral folds generally correspond to universal ribosomal proteins, certain folds that are superfolds in extant genomes such as single α-helices involved in coiled-coils or other helix-helix interfaces (bL32, eL41, and eS30) and Ribosomal protein L24e; Chain: T (bL28 and eL24) are found in kingdom specific ribosomal proteins across the three kingdoms. However, almost all ribosomal protein folds that later evolved into superfolds in modern genomes are shared and thus part of the minimal fold content of the three kingdoms’ ribosomes. Conversely, protein folds specific to each kingdom do not become superfolds in contemporary genomes.

#### 2.2.2. Different Behavior of Kingdom Specific Folds in Bacteria and Archaea/Eukarya

It is important to note that, since the ribosomes of archaea and eukarya have the same ribosomal proteins, with eukaryotic ribosomes containing 12 additional ones, all the ribosomal protein folds that are shared between bacterial and archaeal ribosomes are also shared between bacterial and eukaryotic ribosomes [178]. 

When examining the protein folds specific to ribosomes in each kingdom, several patterns emerge. In bacterial ribosomes, there are 45 distinct topologies, of which 11 are unique to bacteria (Figure 5a). In contrast, archaea and eukaryotes share 14 specific folds that are not found in bacterial ribosomes. Additionally, eukaryotic ribosomes possess five protein folds that are unique to them. This analysis, therefore, reveals a new finding: in the three kingdoms, only approximately half of the kingdom-specific ribosomal proteins have kingdom-specific protein folds (Table 2). For example, among the 18 bacteria-specific ribosomal proteins (Figure 5b), only 9 (bL9, bL17, bL20, bL25, bL35, bL36, bS16, bS18, bS20) exhibit bacteria-specific ribosomal protein folds, while the others (bL12, bL19, bL21, bL27, bL28, bL31, bL32, bL34, bS6) possess folds that are already present in universal ribosomal proteins, although there are two in archaeal and eukaryotic ribosomes. Similarly, out of the 31 archaea- and eukarya-specific ribosomal proteins, only 18 exhibit the archaea- and eukaryotic-specific ribosomal protein folds, and out of the 12 eukarya-specific ribosomal proteins, only 5 have folds that are not present in the ribosomes of the other kingdoms.

Interestingly, the topologies unique to bacterial ribosomes each occur only once within their ribosomal structure, indicating a limited reuse of these specific folds. However, in archaeal and eukaryotic ribosomes, some of the unique protein folds are not only repeated multiple times but are also among the most prevalent. For instance, protein folds such as the N-terminal domain of TfIIb, the UB roll, and the 60S ribosomal protein L30; Chain A are frequently observed in both archaeal and eukaryotic ribosomes.

This difference in the occurrence of specific folds highlights the evolutionary divergence between these kingdoms. In bacterial ribosomes, the uniqueness of certain folds (which is also illustrated by the low bacterial protein/fold ratio—see Table 1) may reflect more specialized functions or bacterial-specific evolutionary constraints. Conversely, the repeated use of specific folds in archaeal and eukaryotic ribosomes suggests a greater degree of functional redundancy and versatility. These reused folds are likely essential for the complex regulatory and structural requirements of archaeal and eukaryotic ribosomes, contributing to their evolutionary success and adaptability.

### 2.3. Imitation Game: Fold Mimicry across Kingdoms

Another interesting point is that, in many situations, folds can interchange to bind to equivalent areas of bacterial and eukaryotic ribosomes. We have systematically selected and compared equivalent and conserved rRNA regions that are bound by different folds in bacterial ribosomes on the one hand, and archaeal or eukaryotic ribosomes on the other. Figure 6 and Table 3 show that, according to the current CATH classification, folds of other categories, in the LSU and SSU, replace six folds and four folds, respectively. This analysis shows, first, that there is a wide variety of ways in which different protein folds can bind to an equivalent rRNA structure. Figure 4 shows, for example, that, in the eS8/bS20 pair, beta-sheets and alpha-helices can be easily interchanged for binding the same rRNA motif and even penetrating deeply into very narrow rRNA crevices. A more detailed analysis shows that the 10 cases listed in Figure 6 differ in the way the folds replace each other to bind to the rRNA.

The eL15/bL28 pair is representative of an extreme situation in which two different folds occupy the same RNA cavity (bounded by the H10, H16, H52, and H79 RNA helices), binding in completely different ways. The interaction surfaces and RNA regions contacted by each of the two proteins are totally different. However, the two folds share three equivalent anchor points: (i) Arg 50 (bL28) and Arg 32 (eL15) point to C2200 (H79), (ii) Lys 3 (bL28) and Lys 103 (eL15) point to G1369 (H52), and (iii) Arg 40 (bL28) and Lys 89 (eL15) are oriented towards U2232 (H75). 

In the eS8/bS20 pair, the situation also shows two proteins of very different folds and sizes binding to a similar region. While eS8 is much larger than bS20 and interacts with a larger surface area of RNA, both proteins enter the same narrow cavity via two different secondary structure elements, helices. 

In the following cases, the different folds seem to precisely target a common RNA interaction surface. eS1, which appears to derive from the duplication of a motif based on three small antiparallel beta-sheets, binds to the same region as the bS6/bS18 pair in the common space they occupy. So, although they have different folds, these proteins will target the same RNA interaction surface in proximity to uS11. eL33 and bL20, which have completely opposite folds in secondary structure content (beta barrel for eL33 and helix bundle for bL20), also concentrate their RNA binding on the same region of helix H41. bS16 occupies the same region as the N-terminal domain of eS4, which appears to have triplicated a similar domain containing a repeat motif of three small anti-parallel beta-sheets. Although presenting different folds, equivalent regions of both proteins interact with the same RNA surface. The pair eL31/bL17 occupies a roughly similar volume and penetrates the same crevasse bordered by helices H47, H61, H 100, and H10, establishing similar interactions despite their different folds. bL17 extends further, however, interacting with helix H57. 

In the following three cases, our analysis leads us to believe that pairs whose folds have not been classified as equivalent nevertheless come from a common ancestor. Thus, it seems likely that the pairs eL21/bL27, eL24/bL19, and eL40/bL36 each have a related fold from a common ancestor. Interestingly, the alpha-helix-rich eS17 occupies the same region as the additional coiled-coil domain of the uS2 protein, and shows surprising convergence by superimposing an almost similar pattern opposite the body/head junction of the SSU. Note, however, that in the eL21/bL27 pair, both of which appear to adopt a small beta barrel, both feature a different-sized N-terminal extension (shorter in eL21) that dips into a common area between LSU helices H38, H81, H85, H86, and H87. The globular domain of eL21, which appears to have rotated relative to bL27, interacts with more or less the same surface of rRNA.

eL24 and bL19 are both located at the interface of the two subunits, and each interacts with the universal protein uL14. Both occupy a remarkably similar space and interact similarly with neighboring rRNA helices H96, H101, and uL14. However, whereas eL24 forms a beta sheet parallel to uL14, bL19 forms an antiparallel sheet. This appears to be due to the relative rotation of the two proteins in the cavity, bringing non-homologous beta-sheets into contact with uL14. 

eL40 and bL36 have similar topologies, containing three small antiparallel beta-sheets. They make very similar contacts in a very narrow cavity bounded by LSU helices H42, H89, H91, and H97. We believe they both originate from a common ancestor and have rotated relative to each other through bacterial/archaeal radiation.

Thus, the permutability or mimicry of folds for the recognition of equivalent rRNA regions, and probably similar functions, shows that processes of molecular convergence have already played an important role, right from the radiation of the three kingdoms.

### 2.4. Fold Evolution/Convergence

Some ribosomal protein folds may have undergone point mutations, insertions, and duplications, leading to the emergence of entirely new folds. Various experimental studies demonstrate that, in certain instances, the genetic routes between completely distinct protein folds can be unexpectedly short [191]. Many proteins exhibit intricate structures composed of multiple layers of secondary elements tightly packed together. These intricate architectures often evolve from simpler, more compact folds. This process involves the elongation of the primary sequence and the subsequent packing of the extended region against the exterior of the ancestral fold [192]. This computational analysis revealed that several ribosomal proteins resemble the minimal motifs of larger protein folds. For example, the protein bL36 exhibits the topology of a Greek key, which is a common elementary motif for different protein folds such as jelly roll and immunoglobulin folds. The repetitive nature of the TIM barrel architecture suggests its evolution from the recombination or duplication of a smaller structure [193]. Analysis revealed that the globular part of eL32 closely matches the minimal structural unit of the TIM barrel, suggesting that eL32 could be its ancestor that underwent recombination or duplication. 

As mentioned earlier, the proteins uL1 and uS2 exhibit minimal RLM [190]. The majority of the big ribosomal proteins consist of two or three domains, with only exceptions being uL4 and uL5. In the CATH system [194], both uL4 and uL5 have their own protein folds, RP l4; Chain A and 50s RP L5; Chain A, respectively. However, a closer look at these protein structures reveals some similarities with already existing protein folds. It was already noted that uL4 is homologous to protein tyrosine phosphatase (PTPase) [195]. Remarkably, its hydrophobic core has four parallel beta-strands which show a strong analogy with minimal RLMs, thus grouping it with uL1 and uS2 (Figure 7). In addition to this, the fact that its beta sheets are parallel suggests some similarity with the RLM. This implies that it can be categorized together with uS2 and the small domain of uL1. Similarly, uL5 exhibits a βαββαβ motif, with the four β-strands forming an antiparallel β-sheet and the α-helices packed on one side, and can be grouped together with uS6, the first domain of uL1, uL6, L7/L12, bL9, uL22, and uL30 [196,197]. This joins the idea of the existence of urfolds and metafolds [89,158,159].

### 2.5. The Inexorable Expansion of Fold Extensions

How do ribosomal protein folds acquire extensions? First, when examining the structured extensions of ribosomal proteins, it has been observed that these extensions typically arise by elongating pre-existing secondary structures within the fold. For instance, ribosomal proteins with β-barrel folds, such as the OB fold, often extend two β-sheets to form a β-hairpin. Similarly, ribosomal proteins composed mainly of α helices tend to lengthen the helices at their N- and C-termini to create extensions. Second, flexible loops can form in various locations: between two sheets, between two helices, or between a sheet and a helix. Notably, these extensions develop without disturbing the overall protein fold.

Analyzing the extensions of universal ribosomal proteins across the three kingdoms revealed that the universal extensions are conserved (Figure 8). There are also certain extensions that are absent in bacteria and are present in both archaea and eukarya and vice versa (Appendix A). For example, the proteins uL3, uL4, uL13, uL18, uL22, uL23, and uL33 in LSU and uS17 in SSU are the most “evolvable” because they have acquired extensions at each evolutionary transition (Figure 9). In contrast, certain proteins such as uL1, uL6, uL10, uL11, and uL15 in the LSU and uS7, uS9, uS10, uS12, uS13, and uS19 do not appear to develop extensions over the course of evolution.

Furthermore, a comparison of archaeal and eukaryotic ribosomal proteins showed that eukaryotic ribosomes have acquired 53 new extensions [87]. The acquisition of extensions can also occur within a single kingdom. For instance, ribosomal protein bS16 typically lacks extensions, but in the *mycobacterium* ribosomes, it features a polypeptide chain ending with a helix at the interaction site. Additionally, structural analysis of ribosomal protein folds in modern genomes identified proteins with the same fold as a lot of the ribosomal proteins, such as uL22 presenting the RP L22; Chain A fold and the ones containing Rossmann fold, Elongation factor Tu, OB, and SH3 domains but without these extensions. More generally, many ribosomal proteins are small β-barrels that tend to be decorated with extensions [89]. These findings do not support the idea that extensions preceded folded proteins [84]. Instead, our analysis indicates that folded proteins emerged first and were later decorated with extensions. This aligns with the broader trend of increasing structural disorder with organism complexity, as the percentage of intrinsically disordered proteins rises from bacteria to archaea to eukaryotes [198,199].

## 3. Discussion

### 3.1. What’s New from the Past?

This study delves into the intriguing usage of protein folds within ribosomes across different kingdoms—bacteria, archaea, and eukaryotes. This analysis shows that fold usages in the ribosomes are similar to those of modern genomes in that they are also part of what might be called the infancy of a power law, and are strongly biased in favor of superfolds. Thus, similar evolutionary principles and mechanisms shape the organization of ancestral nucleo-protein systems and modern proteomes. One of the key findings is the identification of SRFs, which are highly prevalent within ribosomes and appear to have significant functional roles. The OB fold and SH3-type barrels are notable examples of these SRFs. Interestingly, this study finds that some bacterial SRFs have retained their prominence in modern genomes, suggesting their ancient and essential roles. However, despite similarities in the distribution laws, a notable difference was observed. Whereas the superfolds of modern genomes essentially correspond to folds that can accommodate a large number of catalytic functions such as the TIM-barrel or the Rossmann fold, the SRFs are essentially associated with allosteric functions or RNA recognition. In this respect, it is particularly interesting to note that the proto Rossman folds already present in the ribosome in the uL1, uS2, and uL4 proteins are not yet catalytic and have not yet achieved the success of their descendants. This fold, which will undergo a vertiginous expansion in modern genomes due to its huge diversity of catalytic functions, probably began its history in the ribosome due to its ability to recognize RNA and its allosteric functions.

### 3.2. Ancestral Evolutionary Mechanisms with Different Selective Pressure

These differences, therefore, reflect distinct selective pressures on cellular systems and on the ancestral forms of nucleo-protein complexes. Interestingly, a distinct pattern of fold usage is observed in the three kingdom ribosomes, highlighting the evolutionary paths and functions of ribosomal proteins. Figure 8 provides a comprehensive visual summary of the observed patterns, illustrating the distribution of protein folds in the ribosomes of the three kingdoms. Bacterial and eukaryotic ribosomes are clearly distinct in terms of fold usage. The protein-to-fold ratio, which increases from bacteria to eukarya, reflects the evolutionary trend towards more complex and specialized protein functions without a proportional increase in fold diversity. Furthermore, this study highlights the less-differentiated nature of bacterial large subunits (LSU) and small subunits (SSU) compared to those of archaea and eukaryotes. This differentiation in fold content aligns with the broader evolutionary narratives of these kingdoms, where archaea and eukaryotes have developed more specialized and complex ribosomal structures. A notable observation is the preferential use of SH3 domains rather than OB folds in eukaryotic ribosomes. This shift towards a SRF more conducive to allosteric functions probably reflects increased signaling in eukaryotic ribosomes with more complex functions. Transcending the concept of folds also makes it possible to group a large number of ribosomal proteins into only five categories of urfolds or metafolds. This creates continuity in the fold space of the ribosomal protein, which could attest to their ancestral character and common origins (Table 4). For example, grouping into broader classes such as urfolds on the one hand, and categorizing ribosomal proteins by their folds (Figure 5 and Table 2) on the other, provides a fresh look at specific kingdom proteins. A much smaller number of protein families make up the ribosome, and almost half of all specific-kingdom proteins have common origins.

### 3.3. Imitation Games at the Ribosome Age

The observation of the interchangeability of a large number of folds for the recognition of the same ribosomal RNA structures shows that, from the earliest evolutionary stages, evolution already had a large semantic repertoire for performing the same functions. The convergence or mimicry of folds is once again an evolutionary trait found at different evolutionary periods. This shows that evolutionary processes have been surprisingly similar since the first forms of life.

### 3.4. Order First, Disorder Next?

The demonstration that a large proportion of the ribosomal protein’s ancestral folds gradually acquired various categories of extensions during their transitions to the three kingdoms also reveals similar evolutionary mechanisms in ancient nucleo-protein complexes and modern proteomes that are prone to insertion/deletion events. The inexorable expansion of extensions observed throughout ribosome evolution suggests that this trend probably preceded the radiation of the three kingdoms, and thus contributed to the acquisition of universal protein extensions already present in ancestral ribosomes. For example, it is likely that, already in the first evolutionary steps that formed the protoribosomes, the ancestral folds of the uL2, uL3, and uL4 proteins acquired extensions that reach the peptidyl-center or peptide tunnel to enable allostery and signaling between functional centres. These findings support an “order first, disorder next” evolutionary model in which simple but ordered ribosomal proteins preceded the appearance of disordered extensions. This model is also in good agreement with all the data showing that protein disorder increases over the course of evolution and reaches an apex in eukaryotes (Figure 10).

## 4. Materials and Methods

The PDB [184] served as the primary resource for accessing high-resolution crystallographic and cryo-EM ribosome structures representative of bacteria (4Y4P, 4YBB, 4V9H, 7P7Q, 1DD3), archaea (4V6U), and eukarya (4V88, 3J7R, 4UG0). The ribosomal proteins were considered in the context of the whole ribosomes, and some proteins were isolated when needed. Additionally, it provided structures of proteins possessing known folds, facilitating comparative analyses with ribosomal proteins. To categorize protein folds, the CATH database [194], containing hierarchic classification of protein domain structures [115], was utilized. The classification system comprises four main levels: protein class (C), architecture (A), topology (T), and homologous superfamily (H). We used the fold identification search, by sequence and by structure, of the CATH database by isolating the ribosomal proteins from the overall structure of the ribosomes. For identifying proteins with analogous structures to ribosomal proteins whose folds were not directly identified by CATH, Foldseek [200] and Distance matrix alignment (Dali) [201] servers were employed. When a protein with similar structure was identified, its topology was attributed to the ribosomal protein in question. Visualization and structural alignments were performed using PyMOL 2.5.2 [202]. 

## 5. Conclusions

In summary, this comparative analysis of ribosomal protein folds across different kingdoms provides valuable insights into the evolutionary mechanisms and functional adaptations that have shaped modern proteomes. Surprisingly, this study suggests that the evolutionary mechanisms and principles observed in modern genomes, such as the preferential use of folds, convergence, and the acquisition of insertions, were already at work before the radiation of the three kingdoms.

This short paper provides a new look at ribosome evolution by focusing on the protein fold contents and occurrences. The identification of SRFs and their roles underscores the intricate interplay between protein structure and function, offering a deeper understanding of the evolutionary dynamics of ribosomes and their components. These SRFs announce the future power-law distribution observed in extant genomes. OB folds and SH3-type barrels are found to be the most common folds in bacterial and archaeal/eukaryotic ribosomes, respectively. Notably, the Rossmann fold, a prevalent superfold in contemporary proteomes, is not yet dominant in ribosomes. This suggests that different evolutionary constraints other than enzymatic capacities shape the ribosomal protein fold landscape. The observation that different folds can recognize equivalent rRNA structures illustrates an early mechanism of molecular convergence. This comparative analysis deepens our understanding of ribosomal evolution, especially the unique evolutionary trajectories of the large and small subunits, and highlights the intricate interactions between RNA and protein components during the shift from the RNA world to contemporary cellular life. This and our previous works also demonstrate that the gradual acquisition of extensions by simple but ordered folds constitutes an inexorable evolutionary mechanism, which finds its apex in the transition from archaeal ribosomes to eukaryotic ribosomes. This observation supports the idea that simple but structured ribosomal proteins preceded the development of their disordered extensions. Our work also reveals an interesting phenomenon of congruence: the increasing acquisition of extensions involved in the interconnection of functional centres, on the one hand, and the preferential use of SH3 domains conducive to allostery, on the other, show that evolution has made every effort to develop signaling in eukaryotic ribosomes.

## Figures and Tables

**Figure 1 ijms-25-08806-f001:**
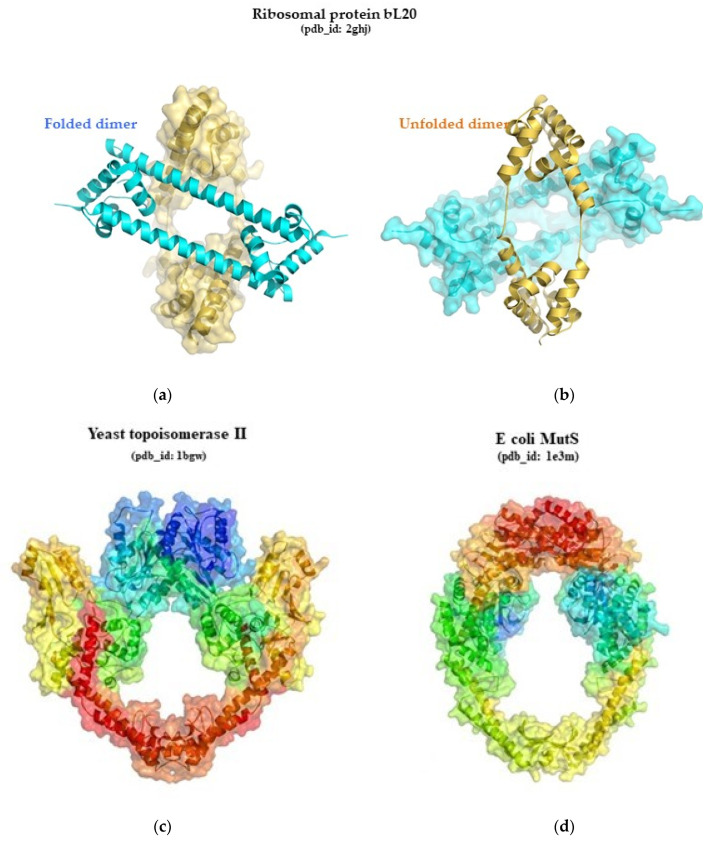
(**a**,**b**): the two folding states of the ribosomal protein bL20 pdb_id: 2ghj; (**c**,**d**): similar rings formed by topoisomerase II (pdb_id: 1bgw) and MutS (pdb_id: 1e3m) with opposite chain polarity (colored with rainbow color code in Pymol (from blue, N-ter to red C-ter).

**Figure 2 ijms-25-08806-f002:**
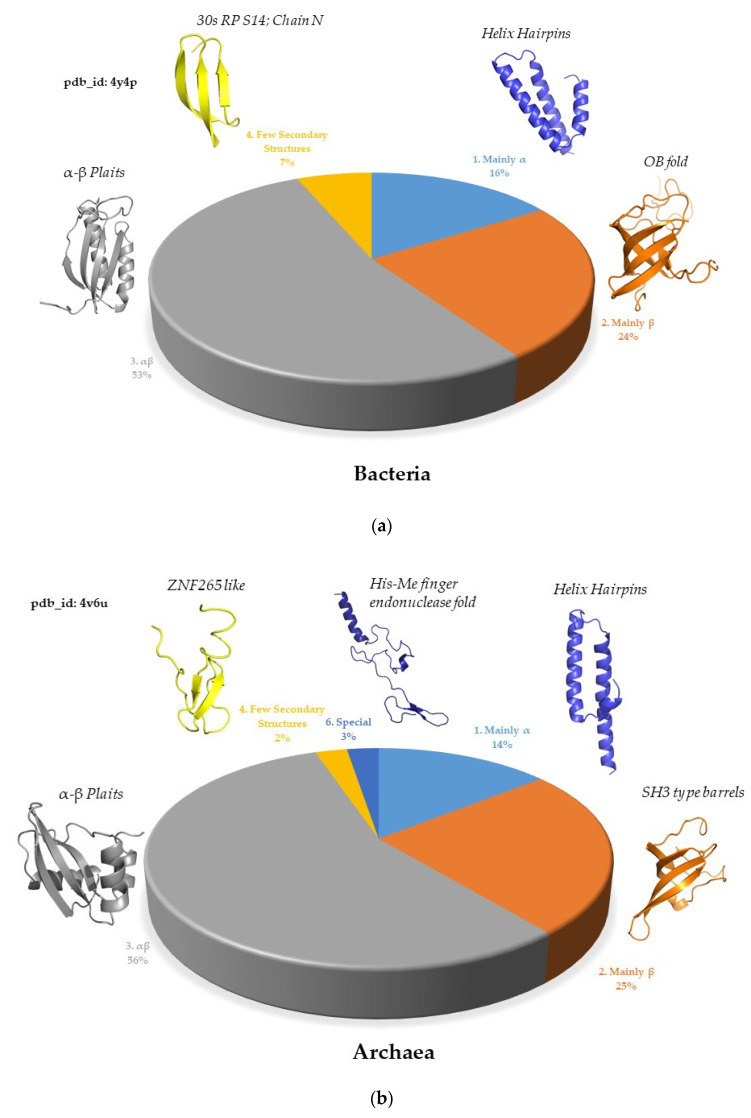
Ribosomal protein fold distribution: (**a**) The distribution by protein fold class in bacterial ribosomes pdb_id: 4y4p; (**b**) the distribution by protein fold class in archaeal ribosomes pdb_id: 4v6u; (**c**) the distribution by protein fold class in eukaryotic ribosomes pdb_id: 4v88; (**d**) the distribution by protein fold class, architecture, and fold in bacterial ribosomes colored by their occurrences; (**e**) the distribution by protein fold class, architecture, and fold in archaeal ribosomes colored by their occurrences; (**f**) the distribution by protein fold class, architecture, and fold in eukaryotic ribosomes colored by their occurrences.

**Figure 3 ijms-25-08806-f003:**
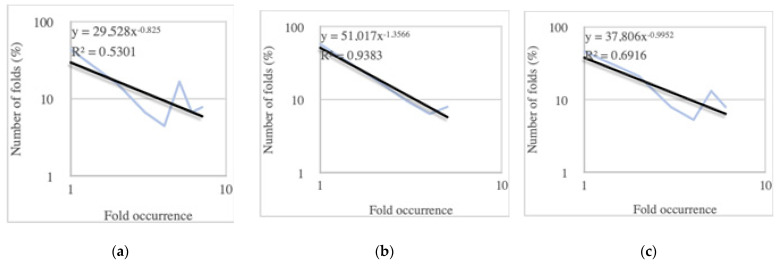
Ribosomal protein fold content represented by counting the occurrences of different protein folds and then grouping those with similar frequencies to display the premises of the power-law distribution, where the number of folds (F) with a specific genomic occurrence (V) follows the equation F = aV^−b^. The vertical axis represents the normalized number of folds relative to the total fold types in the genome on a logarithmic scale, plotted against the occurrence of each fold also on a logarithmic scale in (**a**) bacterial ribosomes; (**b**) archaeal ribosomes; and (**c**) eukaryotic ribosomes [18]. The lines in black are the trend lines and lines in light blue represent the real data. The “super-ribosome folds” of (**d**) bacterial ribosomes; (**e**) of archaeal ribosomes; and (**f**) in eukaryotic ribosomes.

**Figure 4 ijms-25-08806-f004:**
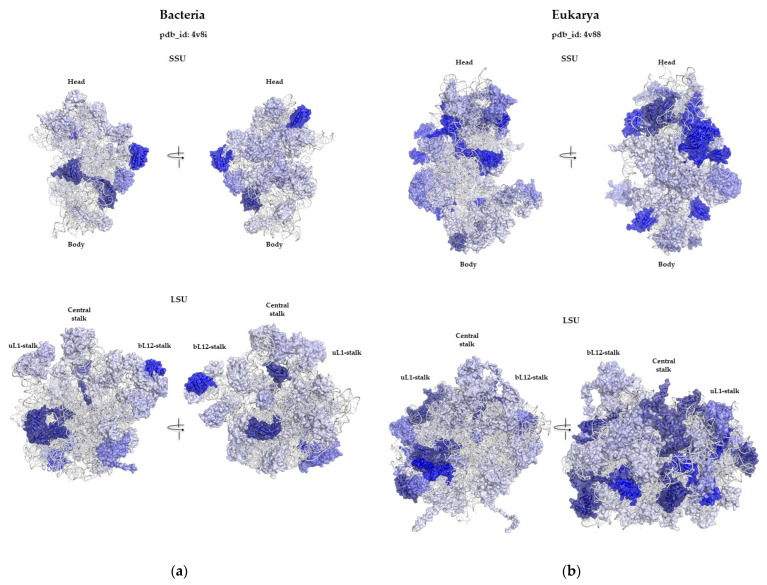
The most prevalent folds colored by their occurrences in the LSU and SSU of (**a**) bacterial pdb_id: 4v8i and (**b**) eukaryotic pdb_id: 4v88 ribosomes. The colors correspond to the colors used in Figure 2d–f.

**Figure 5 ijms-25-08806-f005:**
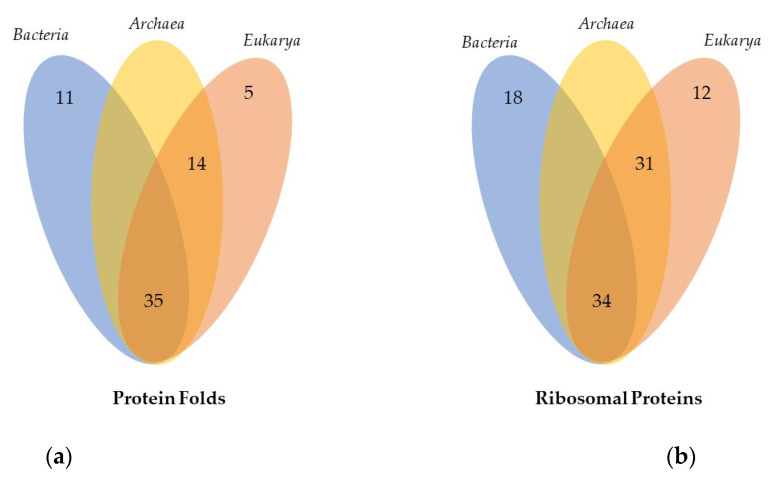
Venn diagrams representing: (**a**) common and specific ribosomal protein folds for each kingdom; (**b**) universal and specific ribosomal proteins for each kingdom.

**Figure 6 ijms-25-08806-f006:**
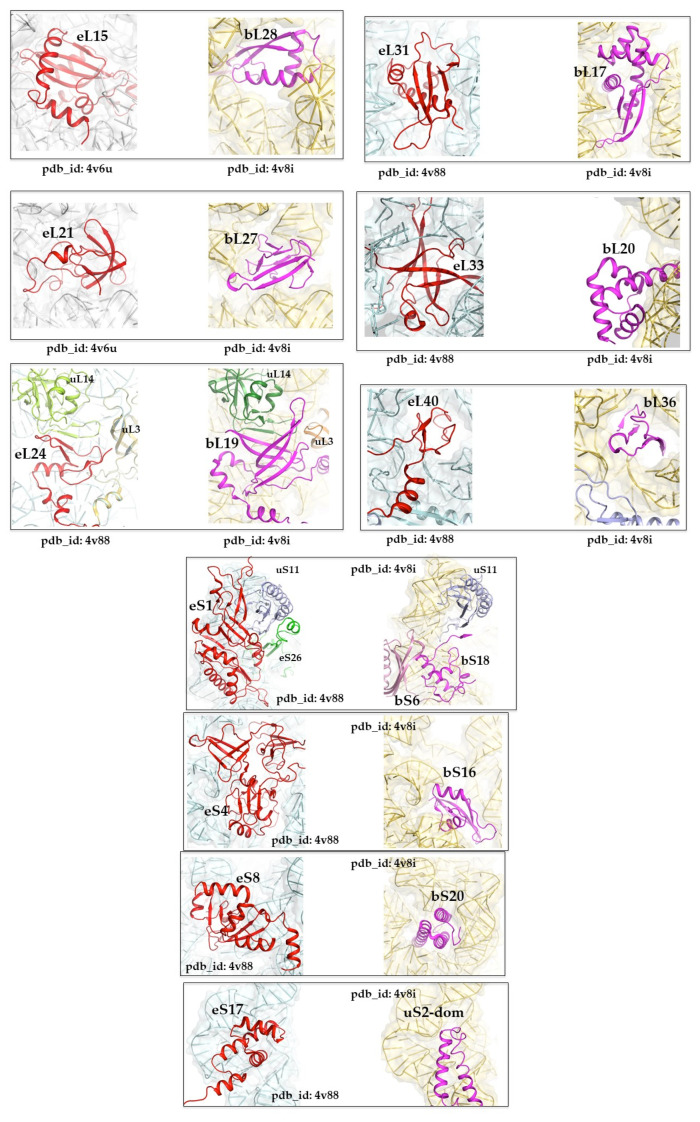
Pairs of different folds that bind equivalent structures in bacterial (pdb_id: 4v8i) and eukaryotic ribosomes (pdb_id: 4v6u and 4v88).

**Figure 7 ijms-25-08806-f007:**
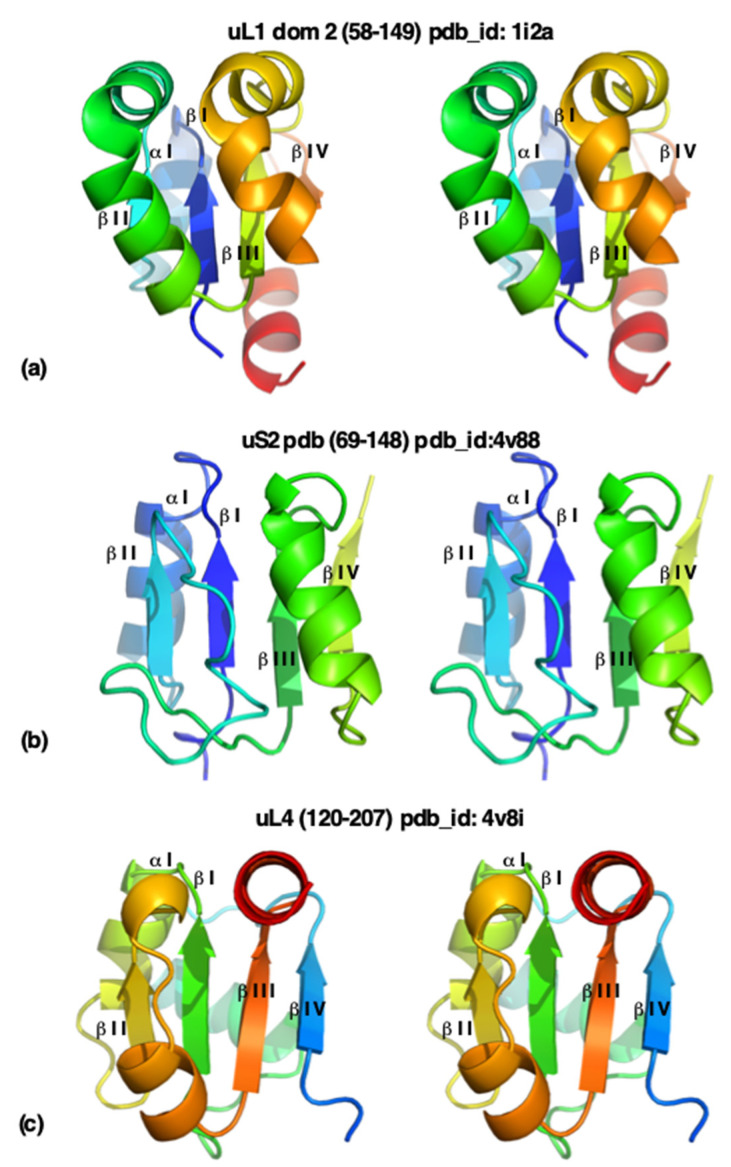
Stereo views of the core of uL1-domain 2 (**a**), of uS2 (**b**), and of uL4 (**c**) colored with the rainbow code in pymol. Similar beta-strands and alpha-helices are indicated.

**Figure 8 ijms-25-08806-f008:**
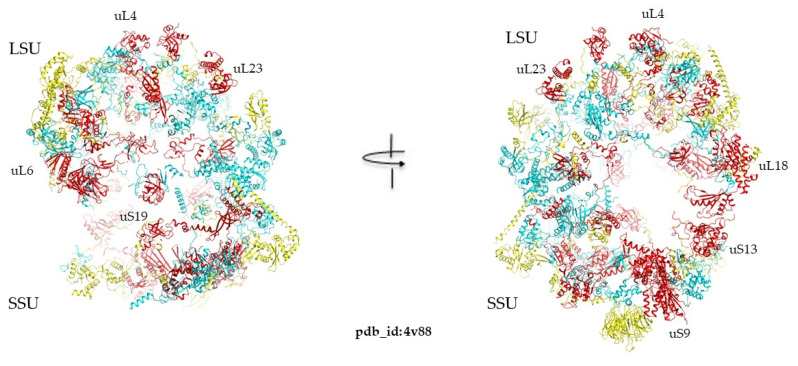
Emergence of extensions: in red: universal proteins and extensions, in blue: proteins and extensions acquired in archaeal and eukaryotic ribosomes, in yellow: proteins and extensions specific to eukaryotic ribosomes pdb_id: 4v88. The rRNA is not represented to highlight the proteins.

**Figure 9 ijms-25-08806-f009:**
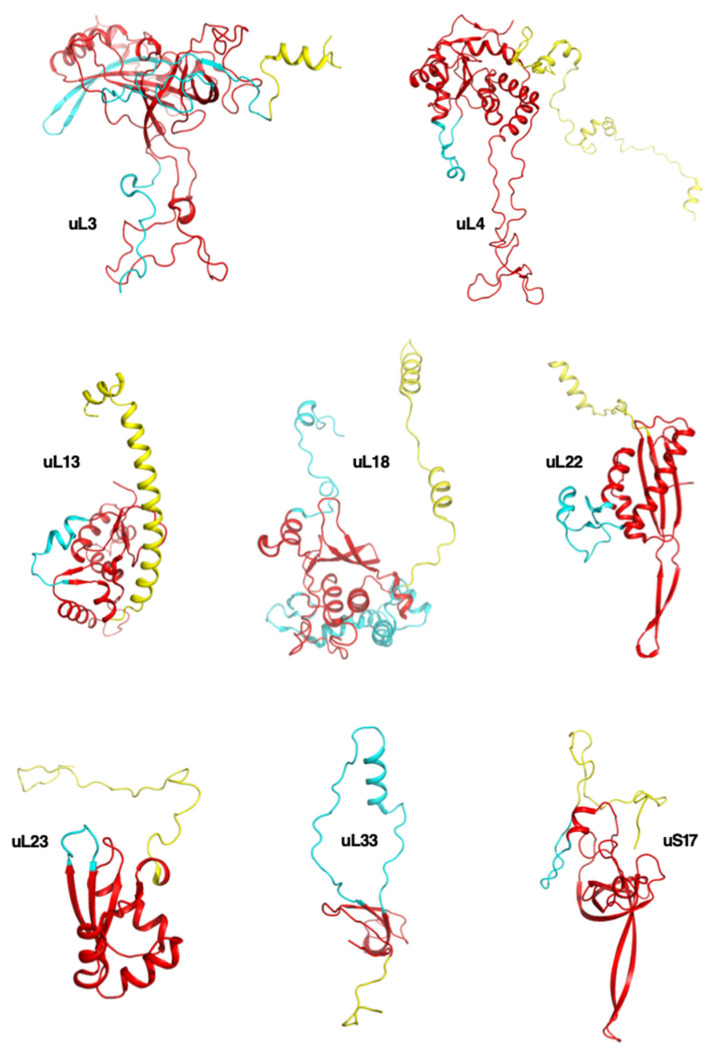
The most evolvable universal ribosomal proteins and their extensions. In red: universal proteins and their universal extensions. In blue: extensions acquired in archaeal ribosomes. In yellow: extensions specific to eukaryotic ribosomes pdb_id: 4v88.

**Figure 10 ijms-25-08806-f010:**
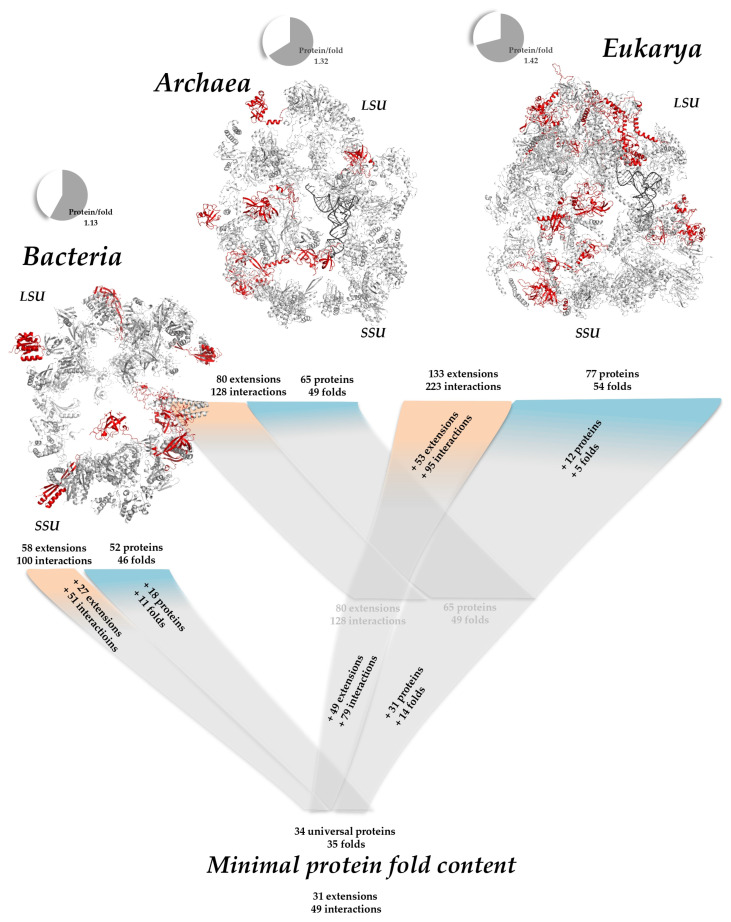
Distribution and evolution of protein folds and their extensions in ribosomes across the three kingdoms [200]. pdb_id: for bacteria 4y4p, for archaea 4v6u, for eukarya 4ug0.

**Table 1 ijms-25-08806-t001:** Comparison of numbers of proteins, folds, and nucleotides in the three kingdoms’ ribosomes and their LSU and SSU.

Kingdom		Proteins	Folds	Protein/Fold Ratio	Nucleotides
Bacteria	LSU	33	32	1.03	3036 ^†^
	SSU	19	20	0.95	1521 ^†^
	Total	52	46 *	1.13	4557 ^†^
Archaea	LSU	39	33	1.18	3175 ^†^
	SSU	26	25	1.04	1495 ^†^
	Total	65	49 *	1.32	4670 ^†^
Eukarya	LSU	44	36	1.22	3675 ^†^
	SSU	33	29	1.14	1800 ^†^
	Total	77	54 *	1.42	5475 ^†^

* The total number of folds in the ribosome was calculated excluding repetitions. ^†^ These are the numbers provided by PDB [184] for 4y4p (bacteria), 4v6u (archaea), and 4v88 (eukarya). The number of nucleotides varies among species.

**Table 2 ijms-25-08806-t002:** Kingdom-specific proteins with kingdom-specific and universal ribosomal protein folds.

	Specific Ribosomal Protein	Specific Ribosomal Protein Fold	Specific Ribosomal Protein	Universal Ribosomal Protein Fold
Bacteria	bL9	3.40.5 RP L9; domain 1 & 3.10.430 RP L9; domain 2	bL12	3.30.1390 RP L30; Chain: A
	bL17	3.90.1030 50s RP L17; Chain A	bL19	2.30.30 SH3 type barrels
	bL20	1.10.1900 c-terminal domain of poly(a) binding domain	bL21	2.40.50 OB fold
	bL25	2.40.240 RP L25; Chain P & 2.170.120 RNApol α Subunit; Chain A, domain 2	bL27	2.40.50 OB fold
	bL35	4.10.410 Factor Xa Inhibitor	bL28	2.30.170 RP L24e; Chain: T
	bL36	2.60.120 Jelly Rolls	bL31	4.10.830 30s RP S14; Chain N
	bS16	3.30.1320 S16 RP; Chain: A	bL32	1.20.5 Single α-helices involved in coiled-coils or other helix-helix interfaces
	bS18	4.10.640 30s RP S18	bL34	1.10.287 Helix Hairpins
	bS20	1.20.58 Methane Monooxygenase Hydroxylase; Chain G, domain 1	bS6	3.30.70 α-β Plaits
Archaea and Eukarya	eL8	3.30.1330 60s RP L30; Chain: A	eL14	2.30.30 SH3 type barrels
	eL15	3.40.1120 RP L15e	eL18	3.100.10 RP L15; Chain K; domain 2
	eL19	1.10.1650 50s RP L19e, Chain O, domain 1	eL21	2.30.30 SH3 type barrels
	eL20	3.10.20 UB roll	eL24	2.30.170 RP L24e; Chain: T
	eL30	3.30.1330 60s RP L30; Chain: A	eL27	2.30.30 SH3 type barrels
	eL31	3.10.440 RP L31e; Chain: W	eL32	3.30.70 α-β Plaits
	eL33	2.40.10 Thrombin, subunit H	eL41	1.20.5 Single α-helices involved in coiled-coils or other helix-helix interfaces
	eL34	6.20.340 His-Me finger endonuclease fold	eS4 (3 folds)	2.30.30 SH3 type barrels & 2.40.50 OB fold & 3.10.290 Structural Genomics Hypothetical 15.5 Kd Protein In mrcA-pckA Intergenic Region; Chain A
	eL37	2.20.25 N-terminal domain TfIIb	eS8	3.10.290 Structural Genomics Hypothetical 15.5 Kd Protein In mrcA-pckA Intergenic Region; Chain A
	eL39	1.10.1620 Atp Synthase ε Chain; Chain: I	eS19	1.10.10 Arc Repressor Mutant, subunit A
	eL40	4.10.1060 ZNF265 like	eS24	3.30.70 α-β Plaits
	eL43	2.20.25 N-terminal domain TfIIb	eS28	2.40.50 OB fold
	eS1	3.10.20 UB roll & 3.30.479 Tetrahydropterin Synthase; Chain A		
	eS6	3.10.20 UB roll		
	eS21	3.30.1230 Hypothetical Cytosolic Protein; Chain A		
	eS27	2.20.25 N-terminal domain TfIIb		
	eS31	6.20.50 N-terminal domain TfIIb		
Eukarya	eL22	3.30.1360 Gyrase A; domain 2	eL6	2.30.30 SH3 type barrels
	eL29	6.10.140 Helix Hairpins	eL13	1.10.10 Arc Repressor Mutant, subunit A
	eL38	3.30.720 Signal recognition particle alu RNA binding heterodimer, srp9/1	eL36	1.10.10 Arc Repressor Mutant, subunit A
	RACK1	2.130.10 Methylamine Dehydrogenase; Chain H	eS10	1.10.10 Arc Repressor Mutant, subunit A
	eS26	3.30.1740 First Zn finger domain of poly(adp-ribose) polymerase-1	eS25	1.10.10 Arc Repressor Mutant, subunit A
			eS30	1.20.5 Single α-helices involved in coiled-coils or other helix-helix interfaces
			Eukarya specific ribosomal protein	Archaea and Eukarya specific ribosomal protein fold
			eS7	2.20.25 N-terminal domain TfIIb
			eS12	3.30.1330 60s RP L30; Chain: A

**Table 3 ijms-25-08806-t003:** Pairs of different proteins and their folds that bind equivalent structures in bacterial (pdb_id: 4v8i) and eukaryotic ribosomes (pdb_id: 4v6u and 4v88).

	Eukarya		Bacteria	
	Protein	CATH	Protein	CATH
LSU	eL15	3.40.1120 RP L15e	bL28	2.30.170 TP L24e; Chain T
	eL21	2.30.30 SH3 type barrels	bL27	2.40.50 OB fold
	eL24	2.30.170 RP L24e; Chain T	bL19	2.30.30 SH3 type barrels
	eL31	3.10.440 RP L31e; Chain W	bL17	3.90.1030 50s RP L17; Chain A
	eL33	2.40.10 Thrombin, subunit H	bL20	1.10.1900 C-terminal domain of poly(a) binding protein
	eL40	4.10.1060 ZNF265 like	bL36	2.60.120 Jelly Rolls
SSU	eS1	3.10.20 UB roll/3.30.479 Tetrahydropterin Synthase; Chain A	bS6/bS18	3.30.70 α-β Plaits/4.10.640 30s RP S18
	eS4	2.30.30 SH3 type barrels	bS16	3.30.1320 S16 RP; Chain A
	eS8	3.10.290 Structural Genomics Hypothetical 15.5Kd Protein In mrcA-pckA Intergenic Region; Chain A	bS20	1.20.58 Methane Monooxygenase Hydroxylase; Chain G, domain 1
	eS17	1.10.60 Diphtheria Toxin Repressor; domain 2	uS2-dom	3.40.50 Rossmann fold

**Table 4 ijms-25-08806-t004:** Categories of identified ribosomal urfolds/metafolds and proteins that exhibit them.

Urfolds/Metafolds	Universal ^1^	B ^2^	AE ^3^	E ^4^
SBBs	uL2, uL3, uL24, uS12, uS17	bL19, bL21, bL27	eL14, eL21, eL24, eL27, eL33, eS4, eS28	eL6
βαββαβ	uL1 (domain1), uL5, uL6, L7/L12, uL10, uL13, uL16, uL23, uS3, uS5 (C terminal domain), uS8, uS9, uS10	bL9 (C terminal domain), bS6	eL30, eL32, eS24	eL22
RLM	uL1 (domain 2), uL4, uS2			
Perpendicularly stacked α helices	uS13, uS2 (coiled coil domain)	bL20	eS17	
3 parallel α helices	uL29, uS15	bL34, bS20	eL19	

^1^ Universal: universal ribosomal proteins. ^2^ B: bacteria-specific ribosomal proteins. ^3^ AE: archaea- and eukarya-specific ribosomal proteins. ^4^ E: eukarya-specific ribosomal proteins.

## Data Availability

Data are contained within the article.

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
