# Peer review of "Protein Fold Usages in Ribosomes: Another Glance to the Past"

_ijms, 2024, doi:10.3390/ijms25168806_

Round 1

Reviewer 1 Report

Comments and Suggestions for Authors

            The manuscript of Tanoz et al reports the analysis of protein fold usage in ribosomes proteins across bacteria, archaea, and eukaryotes- The results highlight the prevalence of certain 'super-ribosome folds' such as the OB fold in bacteria and the SH3 domain in archaea and eukaryotes. In addition, it possible to group a large number of ribosomal proteins into 5 categories of urfolds or metafolds, which could attest to their ancestral character and common origins.

            In my opinion the manuscript is of interest, because it provides new information on how ribosomal proteins evolve. Furthermore, the study highlights that the gradual acquisition of extensions by simple but ordered folds constitutes an inexorable evolutionary mechanism. 

The manuscript can be published after answering some requests listed below.

1- The authors provide information on the databases used but few information on the software used to obtain the results. The authors should increase the information on the analysis software used. Specifically, about their set-up.

2- Did they do any statistical analysis on the data they obtained?

3- Where is the supplementary material? There isn't!

Author Response

Above all, we would like to thank the 3 reviewers for their constructive comments and corrections, which will help to improve the manuscript.

REVIEWER 1

1- The authors provide information on the databases used but few information on the software used to obtain the results. The authors should increase the information on the analysis software used. Specifically, about their set-up.

- The Material and Methods was modified to provide more precise information on the set-up of the computational experiment.

2- Did they do any statistical analysis on the data they obtained?

  • A power-law analysis is provided in the section 3.2. The populations being quite small we provided proportions.

3- Where is the supplementary material? There isn't!

  • Supplementary tables can be found in the Excel file called “Supplementary_Tables.xlsx”

Reviewer 2 Report

Comments and Suggestions for Authors

The manuscript described a comparative study of ribosomal protein folds in bacteria, archaea, and eukarya. Common and specific proteins and folds were investigated to explore their ancestors and understand the evolutionary dynamics of ribosomal proteins. Five urfolds or metafolds were categorized, and ordered ribosomal protein structures preceded the development of disordered extensions were assumed. The study is interesting and meaningful for the investigation of ribosomal protein evolution. It is recommended for the International Journal of Molecular Sciences after further modifications. The suggestions are listed below.

Major concerns:

1.     Three copies of proto-Rossmann folds in ribosomes across all kingdoms, which were described in the abstract, could be collectively described in the sections of general discussion and/or conclusions.

2.     Some of the investigations showed data for both archaea and eukarya, but some showed those independently. To reveal detailed information, it is suggested that the data for archaea, eukarya, and both archaea and eukarya be shown together.

3.     Some of the structures didn’t show the PDB ID. They have to be shown in the figure image and the figure legend, as well as in supplementary files.

4.     Some of the detailed information was unavailable. They have to be provided as supplementary Excel files.

Other concerns:

1.     Section 3, Lines 194-196. The most prevalent protein folds were described as similar to modern genomes. This needs to be described further.

2.     Figure 2. The image resolution needs improvement, especially in Figures 2c and 2d. They could be enlarged to show the details.

3.     Figure 2. The distribution by protein fold class colored by their occurrences could also provide additional supplementary Excel files to show the detailed information.

4.     Figure 2. To reveal the information between archaea and eukarya, it is suggested to show the distribution by protein fold class, also colored by their occurrences, in archaeal and eukaryotic ribosomes separately.

5.     Table 1. Were the total folds from the sum of the LSU and SSU folds, or did it exclude the same folds? Further descriptions are needed in the text and the table footnotes.

6.     Table 1. The information on the availability and the number of nucleotides needs further descriptions in the text and the table footnotes.

7.     Table 1. The information on the number of nucleotides in eukarya showed a range between 4980 and 5280. Further descriptions are needed in the text and the table footnotes.

8.     Figure 2 and Table 1. The data retrieved period has to be indicated in the Materials and Methods. Was the number of nucleotides selected but not all included because they were all integer 0?

9.     Figure 3. The units for the x- and y-axis need to be shown on the plots.

10.  Figure 3. The image resolution needs improvement for Figure 3d-f.

11.  Section 3.3. It was described that the most prevalent folds are largely the same in both subunits (Figure 4a). The types of folds could be indicated.

12.  Figure 4. The types of the most prevalent folds could be described in the legend, and the PDB ID of the exemplified proteins could be indicated.

13.  Section 4.1. The PDB ID of the exemplified proteins could be indicated. In addition, the Structural Genomics Hypothetical 15.5 Kd Protein In mrcA-pckA Intergenic Region could be further described. Was it a Heat Shock Protein 15 kDa (Hsp15)?

14.  Section 4.2. It was described that archaea and eukaryotes share 15 specific folds. But it showed 14 in Figure 5a.

15.  Section 4.2 and Figure 5b. The data shown in Figure 5b was not described in Section 4.2.

16.  Figure 5. The number of ribosomal protein folds and proteins that are unique to archaea could be indicated in Figures 5a and 5b, respectively. In addition, if any, the number of ribosomal protein folds and proteins shared by bacteria and archaea could be indicated in Figures 5a and 5b, respectively.

17.  The data shown in Figure 5 could be shown in a supplementary Excel file.

18.  Figure 6 and Table 2. The PDB ID of the exemplified proteins can be indicated in Figure 6 and Table 2.

19.  Section 6. The universal extensions are conserved in three kingdoms can be described in the text, though Figure 7 was indicated.

20.  Section 6. It was described that there are also certain extensions that are absent in bacteria and are present in both archaea and eukarya and vice versa. The details can be described in the text, though supplementary table 2 was indicated.

21.  Figure 7 and supplementary table 2. The PDB ID of the exemplified proteins can be indicated in Figure 7 and supplementary table 2.

22.  Figure 7. The protein names could be labeled, especially those in blue and yellow.

23.  Figure 7 and supplementary table 2. Were the definitions of colors used in Figure 7 and supplementary table 2 the same? The supplementary table 2 had additional colors.

24.  Figure 8. The image resolution needs improvement.

25.  Figure 8. An additional supplementary Excel file can be provided to show the details of the interactions, extensions, folds, and proteins in the distribution and evolution of protein folds and their extensions in ribosomes across the three kingdoms.

26.  Table 3. An additional column could be added to show the ribosome urfolds and proteins, if any, and describe it in Section 7.

27.  Section 7. Five categories of urfolds or metafolds could be further discussed in Section 7.

28.  Back matter. The title of the two supplementary tables should be described here.

29.  Two supplementary tables should be provided as Excel files. The supplementary table 2a-2d could be arranged in the same Excel file in different sheets. In addition, the definitions of the colors used in the tables need to be indicated.

Author Response

Above all, we would like to thank the 3 reviewers for their constructive comments and corrections, which will help to improve the manuscript.

REVIEWER 2

Major concerns:

  1. Three copies of proto-Rossmann folds in ribosomes across all kingdoms, which were described in the abstract, could be collectively described in the sections of general discussion and/or conclusions.

- This point has been developed in the new version in section 5 and 7 and new figure show the analogy between the minimal Rossmann folds of the domain 1 of uL1, uS2 and uL4 (Figure 7)

  1. Some of the investigations showed data for both archaea and eukarya, but some showed those independently. To reveal detailed information, it is suggested that the data for archaea, eukarya, and both archaea and eukarya be shown together.

- In our previous analysis (Timsit et al. Scientific Reports 2021) it is clearly shown that the eukaryotic ribosome corresponds to archaeal ribosome with some additional proteins and mainly additional extensions and connections. For this reason, we consider them together in some cases, and in the others, they are shown separately to display differences during the transition from archaea to eukarya.

  1. Some of the structures didn’t show the PDB ID. They have to be shown in the figure image and the figure legend, as well as in supplementary files.

- PDB IDs are now indicated.

  1. Some of the detailed information was unavailable. They have to be provided as supplementary Excel files.

- Supplementary Excel files are now provided.

Other concerns:

  1. Section 3, Lines 194-196. The most prevalent protein folds were described as similar to modern genomes. This needs to be described further.

- It was indicated that the protein fold class distribution were similar, not protein folds prevalence. More precise information was added.

  1. Figure 2. The image resolution needs improvement, especially in Figures 2c and 2d. They could be enlarged to show the details.

- The resolution was improved and the figures were enlarged.

  1. Figure 2. The distribution by protein fold class colored by their occurrences could also provide additional supplementary Excel files to show the detailed information.

- The detailed information is indicated in the Supplementary Table 1.

  1. Figure 2. To reveal the information between archaea and eukarya, it is suggested to show the distribution by protein fold class, also colored by their occurrences, in archaeal and eukaryotic ribosomes separately.

- Figure 2c: a pie chart with the distribution by protein fold class in archaeal ribosomes was added to Figure 2.

  1. Table 1. Were the total folds from the sum of the LSU and SSU folds, or did it exclude the same folds? Further descriptions are needed in the text and the table footnotes.

- It is now indicated in the footer of the Table 1 that the total number of folds for each ribosome was calculated excluding repetitions.

  1. Table 1. The information on the availability and the number of nucleotides needs further descriptions in the text and the table footnotes.

- The information about the number of nucleotides is now provided in the table footnote.

  1. Table 1. The information on the number of nucleotides in eukarya showed a range between 4980 and 5280. Further descriptions are needed in the text and the table footnotes.

- A precise number is now given, and it is described in the footnotes.

  1. Figure 2 and Table 1. The data retrieved period has to be indicated in the Materials and Methods. Was the number of nucleotides selected but not all included because they were all integer 0?

- The nucleotide numbers have been found in the pdb files of the corresponding ribosomes. The pdb_id are indicated in the legend

  1. Figure 3. The units for the x- and y-axis need to be shown on the plots.

- The units on the Figure 3 a-c are indicated now.

  1. Figure 3. The image resolution needs improvement for Figure 3d-f.

- The images enlarged and the resolution improved.

  1. Section 3.3. It was described that the most prevalent folds are largely the same in both subunits (Figure 4a). The types of folds could be indicated.

- The types of folds are now indicated and more precise information is provided.

  1. Figure 4. The types of the most prevalent folds could be described in the legend, and the PDB ID of the exemplified proteins could be indicated.

- We used the pdb structures of complete ribosomes and isolated some proteins when needed, so those are the same 4v6u, 4v88, 4y4p, 4ybb that were used for most of the analysis.

  1. Section 4.1. The PDB ID of the exemplified proteins could be indicated. In addition, the Structural Genomics Hypothetical 15.5 Kd Protein In mrcA-pckA Intergenic Region could be further described. Was it a Heat Shock Protein 15 kDa (Hsp15)?

- We used the pdb structures of complete ribosomes and isolated some proteins when needed, so those are the same 4v6u, 4v88, 4y4p, 4ybb that were used for most of the analysis. The ribosomal proteins that exhibit these folds were added.

Structural Genomics Hypothetical 15.5 Kd Protein In mrcA-pckA Intergenic Region is in fact the fold found in Heat Shock Protein 15 kDa (Hsp15), however the details about it the “descendance” of the ribosome were kept for another article with the focus on the relatedness of ribosomal proteins and other proteins that have the same fold. Additionally, we thank you for this remark since it is a mistake, Structural Genomics Hypothetical 15.5 Kd Protein In mrcA-pckA Intergenic Region is present in uS4 which is a universal ribosomal protein.

  1. Section 4.2. It was described that archaea and eukaryotes share 15 specific folds. But it showed 14 in Figure 5a.

- Corrected. It says 14 in the text now.

  1. Section 4.2 and Figure 5b. The data shown in Figure 5b was not described in Section 4.2.

- More description is now provided in the lines 373-381:

In the three kingdoms however only approximately half of the specific ribosomal pro-teins have kingdom specific protein folds. Among the 18 bacteria specific ribosomal proteins (Figure 5b) only 10 exhibit the bacteria specific ribosomal protein folds, while 6 of them possess folds that are already present in universal ribosomal proteins, and 2 in archaeal and eukaryotic ribosomes. Similarly, out of the 31 archaea and eukarya specific ribosomal proteins only 18 exhibit the archaeal and eukaryotic specific ribo-somal protein folds, and out of the 12 eukarya specific ribosomal proteins only 5 have folds that are not present in the ribosomes of the other kingdoms.

  1. Figure 5. The number of ribosomal protein folds and proteins that are unique to archaea could be indicated in Figures 5a and 5b, respectively. In addition, if any, the number of ribosomal protein folds and proteins shared by bacteria and archaea could be indicated in Figures 5a and 5b, respectively.

- As indicated in lines 210-213 and lines 368-371 there are no ribosomal protein folds and ribosomal proteins unique to archaea since all the archaeal ribosomal proteins are shared with eukaryotes. For the same reason all the ribosomal protein folds and ribosomal proteins that are shared among bacteria and eukaryotes are shared among bacteria and archaea.

  1. The data shown in Figure 5 could be shown in a supplementary Excel file.

- The data is shown in the Supplementary Table 1 using the color code and kingdom specific folds are isolated in Supplementary Tables 2 a-b.

  1. Figure 6 and Table 2. The PDB ID of the exemplified proteins can be indicated in Figure 6 and Table 2.

- PDB IDs are now indicated on the figure and in the legends.

  1. Section 6. The universal extensions are conserved in three kingdoms can be described in the text, though Figure 7 was indicated.

-A new figure with conserved and acquired extensions of “evolvable” universal protein has been included in the revised version (Figure 9) and a supplementary table describe the different stages of acquisition of the extensions (supplementary tables 3 a-d). This question is described in detail in a previous paper (ref 87 of the new version)

  1. Section 6. It was described that there are also certain extensions that are absent in bacteria and are present in both archaea and eukarya and vice versa. The details can be described in the text, though supplementary table 2 was indicated.

-The supplementary tables 3a-d and ref 87 provide all the detail of about extension acquisitions

  1. Figure 7 and supplementary table 2. The PDB ID of the exemplified proteins can be indicated in Figure 7 and supplementary table 2.

- We used the PDB structures of whole ribosomes and isolated specific proteins when needed. PDB ID (4v88 corresponding to yeast ribosome) used to create the Figure 7 was added on the figure. The rRNA is not represented for highlighting the proteins.

  1. Figure 7. The protein names could be labeled, especially those in blue and yellow.

- Some of the proteins were labeled to give a general idea about the orientation and not to overcrowd the figure.

  1. Figure 7 and supplementary table 2. Were the definitions of colors used in Figure 7 and supplementary table 2 the same? The supplementary table 2 had additional colors.

- All the colors in the supplementary tables are now defined in the description of the Supplementary Table 1.

  1. Figure 8. The image resolution needs improvement.

- The Figure 8 has a higher resolution now.

  1. Figure 8. An additional supplementary Excel file can be provided to show the details of the interactions, extensions, folds, and proteins in the distribution and evolution of protein folds and their extensions in ribosomes across the three kingdoms.

- All the additional information about the interactions and extensions can be found in our previous publication (Timsit et al. Scientific Reports 2021) which is now cited in the description of the Figure 8.

  1. Table 3. An additional column could be added to show the ribosome urfolds and proteins, if any, and describe it in Section 7.

- Table 3 contains the ribosome urfolds and proteins and this question has been described in section 7 of the revised version

  1. Section 7. Five categories of urfolds or metafolds could be further discussed in Section 7.

-The urfolds and metafolds are discussed in section 7

  1. Back matter. The title of the two supplementary tables should be described here.

- Titles and descriptions are now indicated above every supplementary table.

  1. Two supplementary tables should be provided as Excel files. The supplementary table 2a-2d could be arranged in the same Excel file in different sheets. In addition, the definitions of the colors used in the tables need to be indicated.

- Supplementary tables can be found in the Excel file called “Supplementary_Tabels.xlsx” and the colors were defined.

Reviewer 3 Report

Comments and Suggestions for Authors

The submitted file presents unusual kind of manuscript, something between the original work (although the analysis were quite simple) and review (large number of references) and opinion-article (a lot of general discussion with relation to totally different fields, i.e. literature, arts, etc.). Generally, the work is about the protein fold usages in ribosomes, as the title suggests. It’s quite informative, easy to read, supported with the appropriate references. I would suggest to give it a chance, after carefully considering the requested revisions, listed below.

Major comments:

Lines 70-71, at this point, it would be beneficial to include two-three sentences about the structure and properties of ribosomes, just to remind the readers that are not directly in this fields

Lines 105-117, this sounds more like conclusions…. Consider moving this part at the end.

Lines 118-165, while the Authors describe protein folding in a detailed way, there are no information about the topics like misfolding, prions, proteinopathies. I think the Authors should described them, in the context of the analyzed ribosomal proteins. Are they affected by those conditions? Why? Why not?

Also, the thermodynamics, which is the driving force of whole life, should be included in the discussion. Is folding enthalpy or entropy-driven? Does the conformations with lowest Gibbs free energy really the “desired” one (prions!). Why?

The style of the figures should be polished too, especially Figures 2-4 as some of their context can’t be read.

Lines 207-213, aren’t there really any exceptions to those values?

Figure 3, abc, those equations and their graphical representations don’t look convincing. Those black curves look like straight lines. Maybe another type of functions would be a better representation? It should be properly analyzed from the mathematical point of view, providing the R2 values for each choice.

Lines 545-546, why the Authors haven’t considered the AlphaFold database? There are a lot of ribosomal proteins there that can’t be found in RCSB PDB.

Minor issues:

References 136-138, this is quite unusual to cite MS Excel or MS Word or Python. Those are well known tools and their owners don’t require this. I recommend removing those references.

Line 22, it should be “urfolds”

Table 3, “B”, “AE”, “E” should be explained in a footnote

Line 89, “LUCA” should be explained

Lines 237-262, this text should be justified

Author Response

Above all, we would like to thank the 3 reviewers for their constructive comments and corrections, which will help to improve the manuscript.

REVIEWER 3

Lines 70-71, at this point, it would be beneficial to include two-three sentences about the structure and properties of ribosomes, just to remind the readers that are not directly in this fields

-The revised version describes the ribosome functions and structure in the corresponding section

Lines 105-117, this sounds more like conclusions…. Consider moving this part at the end.

-it has been removed from the introduction

Lines 118-165, while the Authors describe protein folding in a detailed way, there are no information about the topics like misfolding, prions, proteinopathies. I think the Authors should described them, in the context of the analyzed ribosomal proteins. Are they affected by those conditions? Why? Why not?

-Thank you for stimulating the exploration of this interesting topic. The description of prions, prion diseases and misfolding as well as the role of RNA and ribosomes in misfolding process has been included in the new version

Also, the thermodynamics, which is the driving force of whole life, should be included in the discussion. Is folding enthalpy or entropy-driven? Does the conformations with lowest Gibbs free energy really the “desired” one (prions!). Why?

-References and description of the folding processes (thermodynamic and kinetic) has been included in the new version

The style of the figures should be polished too, especially Figures 2-4 as some of their context can’t be read.

-This problem has been solved in the revised version

Lines 207-213, aren’t there really any exceptions to those values?

  • It is now indicated in the lines 224-227 that:

It is possible to observe slight variations in these numbers because the quantity of pro-teins can differ not only among kingdoms but also among species, among different strains of the same bacterial species, and even between different tissues within the same organism in multicellular organisms [118–122].

Figure 3, abc, those equations and their graphical representations don’t look convincing. Those black curves look like straight lines. Maybe another type of functions would be a better representation? It should be properly analyzed from the mathematical point of view, providing the R2 values for each choice.

  • The R² values were added and it is not indicated in the lines 281-283 that

“When the numbers of folds in terms of percentages is plotted against the fold occur-rences on a log-log graph, a straight line can be observed which indicates the presence of a power law relationship.”

The graphs were also further discussed in the same section 3.2.

Lines 545-546, why the Authors haven’t considered the AlphaFold database? There are a lot of ribosomal proteins there that can’t be found in RCSB PDB.

  • The ribosomal proteins were analyzed in the context of a ribonucleic complex (ribosome) using cryo-EM structures of complete ribosomes downloaded from PDB. AlphaFold does not contain complete ribosomal models yet.

Minor issues:

References 136-138, this is quite unusual to cite MS Excel or MS Word or Python. Those are well known tools and their owners don’t require this. I recommend removing those references.

  • Removed

Line 22, it should be “urfolds”

  • Changed

Table 3, “B”, “AE”, “E” should be explained in a footnote

  • Explained

Line 89, “LUCA” should be explained

  • This is explained and a reference has been added

Lines 237-262, this text should be justified

-The text has been justified in the revised version

Round 2

Reviewer 3 Report

Comments and Suggestions for Authors

The Authors have revised and improved their manuscript. Current version can be accepted.